# Patient Blood Management in Microsurgical Procedures for Reconstructive Surgery

**DOI:** 10.3390/diagnostics13172758

**Published:** 2023-08-25

**Authors:** Maria Beatrice Rondinelli, Luca Paolo Weltert, Giovanni Ruocco, Matteo Ornelli, Pietro Francesco Delle Femmine, Alessandro De Rosa, Luca Pierelli, Nicola Felici

**Affiliations:** 1Department of Transfusion Medicine, San Camillo-Forlanini Hospital, 00152 Rome, Italyluca.pierelli@uniroma1.it (L.P.); 2Department of Medical Statistics, Saint Camillus International University of Health and Medical Sciences (UniCamillus), 00131 Rome, Italy; 3Department of Reconstructive Surgery, San Camillo-Forlanini Hospital, 00152 Rome, Italyfelicinicola.md@gmail.com (N.F.); 4Department of Experimental Medicine, La Sapienza University, 00161 Rome, Italy

**Keywords:** free tissue transfer, microsurgical procedures, reconstructive surgery

## Abstract

**Introduction:** The main purpose of reconstructive surgery (RS) is to restore the integrity of soft tissues damaged by trauma, surgery, congenital deformity, burns, or infection. Microsurgical techniques consist of harvesting tissues that are separated from the vascular sources of the donor site and anastomosed to the vessels of the recipient site. In these procedures, there are some preoperative modifiable factors that have the potential to influence the outcome of the flap transfer and its anastomosis. The management of anemia, which is always present in the postoperative period and plays a decisive role in the implantation of the flap, covers significant importance, and is associated with clinical and laboratory settings of chronic inflammation. **Methods:** Chronic inflammatory anemia (ACD) is a constant condition in patients who have undergone RS and correlates with the perfusion of the free flap. The aim of this treatment protocol is to reduce the transfusion rate by maintaining both a good organ perfusion and correction of the patient’s anemic state. From January 2017 to September 2019, we studied 16 patients (16 males, mean age 38 years) who underwent microsurgical procedures for RS. Their hemoglobin (Hb) levels, corpuscular indexes, transferrin saturation (TSAT) ferritin concentrations and creatinine clearance were measured the first day after surgery (T0), after the first week (T1), and after five weeks (T2). At T0, all the patients showed low hemoglobin levels (average 7.4 g/dL, STD 0.71 range 6.2–7.4 g dL^−1^), with an MCV of 72, MCH of 28, MCHC of 33, RDW of 16, serum iron of 35, ferritin of 28, Ret% of 1.36, TRF of 277, creatinine clearance of 119 and high ferritin levels (range 320–560 ng mL^−1^) with TSAT less than 20%. All the patients were assessed for their clinical status, medical history and comorbidities before the beginning of the therapy. **Results:** A collaboration between the two departments (Department of Transfusion Medicine and Department of Reconstructive Surgery) resulted in the application of a therapeutic protocol with erythropoietic stimulating agents (ESAs) (Binocrit 6000 UI/week) and intravenous iron every other day, starting the second day after surgery. Thirteen patients received ESAs and FCM (ferric carboxymaltose, 500–1000 mg per session), three patients received ESAs and iron gluconate (one vial every other day). No patients received blood transfusions. No side effects were observed, and most importantly, no limb or flap rejection occurred. **Conclusions:** Preliminary data from our protocol show an optimal therapeutic response, notwithstanding the very limited scientific literature and data available in this specific surgical field. The enrollment of further patients will allow us to validate this therapeutic protocol with statistically sound data.

## 1. Introduction

Patient blood management (PBM) is defined as the timely application of evidence-based medical and surgical concepts designed to manage anemia, optimize hemostasis, and minimize blood loss in order to improve patient outcomes [1].

PBM encompasses all aspects of medical care, including patient evaluation, clinical management surrounding disease management, and the appropriate therapeutic decision-making process, including the application of appropriate agents with best indications, as well as identifying prevention to improve outcomes and quality of life [2]. As a result, one of many outcomes attributed to PBM is the reduction of the need for allogeneic blood transfusions and reduced health-care costs, while ensuring that blood components are available for the patients who need them [3].

The three-pillar matrix of patient blood management (Shander et al., 2012 [1]) is a model of evidence-based approaches, including: [4] the optimization of the patient’s endogenous red cell mass, [5] the control of all blood loss, and [6] the harnessing and optimization of patient-specific physiological tolerance of anemia while administering the best therapy and adopting restrictive transfusion thresholds. PBM identifies patients who are planned for surgery and are at risk of transfusion, providing a management plan aimed at reducing or eliminating the need for allogeneic transfusion, thus reducing its inherent risks, inventory pressures, and escalating costs associated with transfusion [7].

Sideropenic anemia is the most common disease in the world’s population, accounting for approximately 45% of patients who are candidates for elective surgery [8,9]. Preoperative clinical assessment of the patient and diagnosis of an anemic state and its therapeutic resolution is of paramount importance for a care pathway with reduced clinical risks and pharmacological interference [9].

Surgery of any kind is a major cause of blood mass loss, which is directly proportional to the degree of invasiveness [10]. Reconstructive surgery provides restoration of the integrity of soft tissues damaged by trauma, surgery, congenital malformation, burns, or infections. Microsurgery has a fundamental role in reconstructive surgery (RS) and is based on the use of free tissue transfer (FTT). Microsurgical reconstruction techniques consist of harvesting tissues that are separated from the native vascular sources of the donor site, then transferring them to a recipient site and revascularizing them by anastomosing them to the vessels of the recipient site [11].

In these procedures, flaps perfusion and settling on the new site are crucial, and there are some preoperative editable factors that have the potential to affect the flap transfer’s outcomes and their anastomoses patency. Some of these factors are ascribable to the time of the surgery, such as the duration of surgery, the trophic conditions of the injured tissues, the local blood perfusion, and the anaesthesiologic procedures, which have a main role in the control of the hemodynamic stability. Among these general factors, anemia is one well-recognized leading aspect [12].

For this surgical setting, it is essential to manage the anemia that generally occurs in the postoperative period, which plays a decisive role in the implantation of the flap and it is associated with a clinical and laboratory setting of chronic inflammation [13]. The application of patient blood management principles, including preoperative optimization, meticulous intraoperative hemostasis, and innovative transfusion alternatives, can mitigate the risk of complications associated with transfusions [14]. Techniques like intraoperative cell salvage, where the shed blood is collected, processed, and reinfused, minimize the need for allogeneic blood transfusions and reduce immunosuppression [15]. Additionally, the use of advanced monitoring technologies, such as near-infrared spectroscopy, can provide real-time feedback on tissue oxygenation and guide interventions to maintain optimal perfusion.

In summary, in patients undergoing RS, ACD poses a continuous challenge to tissue oxygenation and flap viability [16]. By adopting a proactive approach that focuses on managing ACD, minimizing blood hyperviscosity, and reducing the immunosuppressive effects associated with transfusions, clinicians can optimize tissue oxyphoresis and ultimately enhance patient outcomes [17,18]. 

## 2. Methods

Anemia is a constant finding in patients who have undergone microsurgery. Preoperative anemia with value of haematocrit lower than 30% and of haemoglobin lower than 10 g/dL has been reported to compromise the perfusion of a free flap and to result in flap necrosis (Hill et al., 2012) [19,20,21]. In a previous study by Clark et al., 2007 [22], a strong emphasis was placed on maintaining the level of Hct > 30% and of Hb > 10 g/dL before and after microsurgical procedures to reduce the risk of negative outcomes [23]. This risk is enhanced due to the flap transfer itself, where the anastomosed vessels are perfused at different blood pressure values, usually lower than that of the donor site. For this reason, optimal blood oxygenation is necessary to balance this transitory haemodynamic subversion [24]. Nevertheless, there are clinical studies that do not confirm the results showed by Hill et al. and Mlodinow et al. [21,25,26,27], so the correlation between anemia and the outcomes of microsurgical procedures is still controversial. 

Chronic inflammatory anemia (ACD) is a prevalent and persistent condition observed in patients undergoing RS, and its impact on tissue perfusion, particularly in relation to free flap viability, is significant [28]. Maintenance of optimal tissue oxygenation is of paramount importance, as it directly influences the healing process and overall surgical outcomes. Clinicians can mitigate the immunosuppressive effects associated with transfusions and improve tissue oxygenation by preventing blood hyperviscosity and minimizing the need for blood transfusions [29].

ACD, characterized by impaired iron utilization and reduced red blood cell production, poses a challenge in achieving adequate tissue oxygenation during RS [14,30]. Insufficient tissue perfusion due to chronic anemia compromises the viability of free flaps, which rely on robust blood supply for successful integration. Thus, adopting a comprehensive approach to manage ACD in this patient population is imperative [31].

In order to optimize tissue oxygen delivery, it is crucial to implement strategies that minimize blood hyperviscosity and reduce the reliance on blood transfusions [21]. Addressing the underlying causes of ACD, such as inflammation or chronic disease processes, through targeted therapies can help improve erythropoiesis and alleviate anemia. Additionally, judicious use of iron supplementation, in combination with other erythropoietic agents, may be employed to enhance red blood cell production and restore hemoglobin levels [22].

In order to contribute to filling this gap in evidence, we conducted a prospective observational study which aimed to reduce the transfusion rate as primary endpoint and to avoid limb rejection as secondary endpoint. From January 2017 to September 2019, we evaluated 16 patients (16 males, average age of 33 years) who underwent microsurgical procedures for FTT. In all cases, the recipient site was the lower limb, and all the patients were experiencing a single and first trauma setting. All the enrolled patients presented a loss of substance of soft tissues with bone exposure due to an open injury of the limb, which was repaired with an anterolateral thigh flap. 

The patients’ hemoglobin (Hb) levels, corpuscular indexes, transferrin saturation (TSAT), ferritin concentrations, and creatinine clearance were measured the first day after surgery (T0), after the first week (T1), and after five weeks (T2). All the patients were evaluated for their iron balance with the transferrin saturation percentage. Thirteen patients had a percentage lower than 20%, and three patients had TSAT% higher than 20%. The renal function and creatinine clearance levels were normal for all the patients considered [27].

We assessed all the patients for their clinical condition, history, and comorbidities before starting therapy. The continuous variables were expressed the mean and standard deviation, while the categorical variables were expressed as percentages. Tabular views of the baseline characteristics, iron metabolism, and red blood cells parameters are reported in Table 1 and Table 2.

## 3. Results

The collaboration between the two departments (i.e., the Department of Transfusion Medicine and the Department of Reconstructive Surgery) resulted in the application of a therapeutic protocol with erythropoietic stimulating agents (ESAs) (Binocrit 6000 UI/week) and intravenous iron every other day, starting from the second day after surgery [32]. Thirteen patients received ESAs and ferric carboxymaltose (FCM) (500–1000 mg per session), and three patients received ESAs and iron gluconate (1 vial every other day). The difference in the iron therapy used was correlated with the percentage of transferrin saturation, which was above 20% in three patients. The combination of the two treatments chosen, in line with the treatment adopted by a hospital care protocol and formalized by the health management, had the aim of enhancing erythropoietic regeneration and normalizing the martial balance.

At T0, all the patients showed low hemoglobin levels (average 7.4 g/dL, STD 0.71 range 6.2–11.4 g/dL), with an MCV of 72 fL, MCH of 28 pg, MCHC of 33 g/dL, RDW of 16%, serum iron of 35 mcg/dL, ferritin of 28 ng/mL, Ret% of 1.36, TRF of 277 mg/dL, and creatinine clearance of 119 mL/min. The variations in the laboratory parameters from T0 (baseline) to T1 (after the first week) and T2 (after five weeks) are described in Table 3. All the patients were studied for their immunotransfusional aspects (blood group, direct and indirect Coombs test) on the first day of hospitalization. None of the patients received blood units during their hospital stay. No side effects were observed, and above all, there was no rejection of the limb or flap. Cell salvage techniques were not applied during surgery, as the bleeding was surgically controlled.

After assessing the normality using the Shapiro–Wilk test, a null hypothesis was formulated, stating no actual difference in the levels of hemoglobin, ferritin, and TSAT values between timepoints. A two-tailed homoschedastic dependent samples *t*-test was then performed to reject the hypothesis. The results are shown in Table 3 and Table 4.

No immediate surgery-related complications occurred. In one case, superficial necrosis of the flap occurred due to venous congestion. Debridement of the necrotic tissues was performed, and after three weeks of delivering negative pressure wound therapy, the superficial layer of the flap was successfully covered with a skin graft [33,34].

## 4. Discussion

The application of PBM strategies represents an evidence-based best practice in several surgical settings. The reason for this lies in the synergy between reducing healthcare risks, improving outcomes, and health economic savings [35].

Patient blood management (PBM) in surgical settings is an integral aspect of modern healthcare, emphasizing optimal utilization of blood products while prioritizing patient safety and outcomes. PBM represents a state-of-the-art approach that recognizes the potential risks associated with transfusions and aims to minimize them through a comprehensive and evidence-based strategy [36].

The importance of PBM lies in its ability to enhance patient care across the perioperative continuum. By employing preoperative anemia management, judicious use of blood transfusions, and effective bleeding control, PBM strategies reduce the need for transfusions, thereby minimizing the associated risks and complications. This approach not only improves patient safety but also reduces healthcare costs and enhances overall surgical outcomes [37].

The cornerstone of PBM is an individualized patient assessment, considering factors such as age, comorbidities, and surgical complexity. Preoperative optimization through the correction of anemia, nutritional support, and iron supplementation can significantly reduce the need for transfusions. Furthermore, employing minimally invasive surgical techniques and utilizing hemostatic agents during surgery aids in controlling bleeding and reducing blood loss [38].

An essential component of PBM is the concept of a patient blood management team, comprising various healthcare professionals collaborating to implement tailored strategies. This interdisciplinary approach ensures effective communication, knowledge sharing, and adherence to evidence-based guidelines. Moreover, ongoing education and training programs for healthcare providers play a vital role in promoting awareness and implementing best practices related to PBM [39].

The state of the art in PBM includes advanced laboratory testing, such as thromboelastography and point-of-care coagulation monitoring, which enable real-time assessment of the coagulation status and guide transfusion decisions. Additionally, the use of cell salvage techniques, where the shed blood is collected, processed, and reinfused back into the patient, reduces reliance on allogeneic blood transfusions.

In conclusion, patient blood management in surgical settings represents an important paradigm shift in healthcare, focusing on individualized patient care, optimizing blood utilization, and minimizing transfusion-related risks. By implementing evidence-based strategies, fostering interdisciplinary collaboration, and embracing technological advancements, PBM improves patient outcomes, enhances safety, and ensures efficient resource allocation. Embracing PBM as a standard of care in surgical settings can lead to significant advancements in healthcare and ultimately benefit patients worldwide [40].

Reconstructive surgery is a relatively innovative application of PBM strategies. It is fundamental for preserving both the surgical reconstruction and the ongoing oxyphoresis of the FTT. Wound contamination in severe open limb trauma often leads to deep infection. The strategy employed in such cases involves early debridement of the wound, combined with intensive and repeated washing and repair of the loss of substance using FTT.

In patients with open and contaminated trauma, it is quintessential to perform infection control through irrigation and debridement within the first hours after trauma and to cover residual loss of substance with vascularized flaps. Flap survival depends on optimal tissue perfusion and oxygenation; in this regard, anemia plays a fundamental role. Hematocrit levels should be maintained above 30 percent to minimize the number of postoperative transfusions. There is also evidence that blood transfusions are associated with both infectious and noninfectious complications [41].

In this setting, preoperative anemia has a dual pathogenesis, both due to post-surgical inflammation and perioperative bleeding and infection. Notably, inflammation impedes iron turnover through a blockade of the regulatory hormone hepcidin, resulting in a condition of chronic inflammatory anemia. Thus, these patients have multifactorial anemia with a hyperferritinemic status.

Allogeneic blood transfusion often represents a life-saving therapy, particularly in patients with acute bleeding. However, the literature on hemovigilance systems exposes several associated risks with this therapy, primarily including immunosuppressive and blood hyperviscosity as factors that could lead to the rejection of the reconstructed flap, resulting in therapeutic failure.

Scientific evidence has allowed for the use of the described therapeutic protocol involving the use of biosimilars to bypass the hepcidin-induced iron blockade and to stimulate erythropoiesis. The therapeutic use of intravenous iron assists in the action of erythrocyte reproductive stimulation.

The results achieved through the implementation of this therapy have consistently demonstrated a notable and progressive improvement in various erythrocyte parameters. These positive outcomes include a steady increase in red blood cell count, hematocrit, and hemoglobin levels, collectively indicating an enhanced oxygen-carrying capacity within the bloodstream. By maintaining optimal levels of hemoglobin, a key component of erythrocytes responsible for binding and transporting oxygen, this therapy ensures the preservation of optimal tissue oxyphoresis.

The therapy’s effectiveness is further underscored by its ability to sustain the desired hemoglobin levels over time. This maintenance of an adequate hemoglobin concentration plays a critical role in facilitating the efficient delivery of oxygen to tissues throughout the body. As oxygen is indispensable for cellular metabolism and tissue health, ensuring optimal tissue oxygenation is of paramount importance in supporting physiological processes, promoting wound healing, and minimizing the risk of complications.

The observed progressive increase in erythrocyte parameters reflects the therapy’s impact on improving red blood cell production, optimizing iron utilization, or enhancing erythropoiesis, depending on the underlying cause of the anemia. By addressing the root cause of the anemia, this therapy effectively restores erythrocyte function, leading to an improved oxygen-carrying capacity and overall tissue oxygenation.

The consistent elevation in erythrocyte parameters also suggests that this therapy promotes the long-term sustainability of improved blood parameters, thereby reducing the likelihood of relapses or fluctuations in hemoglobin levels. This stability in erythrocyte parameters is crucial for maintaining sustained tissue oxyphoresis and ensuring the ongoing delivery of oxygen to organs and tissues, supporting their normal physiological functions.

Overall, the results obtained using this therapy show its efficacy in progressively enhancing erythrocyte parameters, sustaining optimal hemoglobin levels, and optimizing tissue oxyphoresis. By improving oxygen delivery to the tissues, this therapy holds significant promise in promoting overall patient well-being, facilitating efficient wound healing, and enhancing physiological processes that rely on adequate tissue oxygenation [42]. All the patients were evaluated for transfusion tests, but none of the patients received blood or other blood component transfusions, decreasing the risks of flap rejection and infection. No patients in the study experienced flap rejection, and their hematological parameters normalized after five weeks.

The preliminary data from our protocol show an optimal therapeutic response in a surgical setting with limited scientific literature available. The enrollment of other patients will allow us to validate this therapeutic protocol with statistically significant data.

## 5. Conclusions

Patient blood management shifts the focus from the product to the patient and considers the patient’s blood as a resource that should be stored and managed appropriately as a standard of care. The fundamental objectives of PBM are to improve clinical patient safety and prevent avoidable transfusion through timely management of all modifiable risk factors, consequently reducing management costs.

Considerable scientific evidence, clinical trials, and guidelines emphasize and encourage the application of PBM pathways. However, to date, the international and national application of these pathways is uneven and sometimes disorganized, resulting in ineffectiveness.

This course of treatment represents an innovative modality within this surgical setting. Its implementation will not only ensure the definition of new endpoints but also overcome decisive clinical phases for these patients.

## Figures and Tables

**Table 1 diagnostics-13-02758-t001:** Baseline red blood cell parameters.

Value	Mean	St Dev
HGB (g/dL)	7.4 g/dL	0.71
HCT (%)	24.2%	2.14
MCV (fL)	72.06 fL	1.48
MCH (pg)	27.94 pg	1.29
MCHC (g/dL)	33.1 g/dL	1.27
RDW (%)	15.93%	7.27

**Table 2 diagnostics-13-02758-t002:** Baseline iron metabolism and kidney function.

Value	Mean	St Dev
Serum iron (mcg/dL)	35 mcg/dL	15
FER (ng/mL)	28.56 ng/mL	20.08
RET (%)	1.36%	0.75
TRF (mg/dL)	277.44 mg/dL	64.76
Creatinine clearance (mL/min)	117.5 mL/min	8.49

**Table 3 diagnostics-13-02758-t003:** Average erythrocyte parameters at different study timepoints. T0 baseline, T1 after the first week, T2 after five weeks.

Parameter	T0	T1	T2	*p* T0 vs. T1	*p* T1 vs. T2	*p* T0 vs. T2
Haemoglobin (g/dL)	7.4	8.43	10.49	<0.001	<0.001	<0.001
Hematocrit (%)	24.2	28.31	36.69	<0.001	<0.001	<0.001
MCV(fL)	72.06	76.44	83.13	<0.001	<0.001	<0.001
MCH (pg)	27.94	31	33.63	<0.001	<0.001	<0.001
MCHC (g/dL)	33.1	32.96	33.64	0.70	0.20	0.001
RDW (%)	15.93	19.92	23.81	0.45	0.001	<0.001

**Table 4 diagnostics-13-02758-t004:** Iron metabolism and kidney function at different study timepoints.

Parameter	T0	T1	T2	*p* T0 vs. T1	*p* T1 vs. T2	*p* T0 vs. T2
Serum Iron (mcg/dL)	35	49.75	68.38	<0.001	<0.001	<0.001
Ferritin (ng/mL)	28.56	120.63	150	<0.001	<0.001	0.003
Reticolocites (%)	1.36	1.66	2.24	0.10	<0.001	0.003
TRF (mg/dL)	277.44	271.94	145.94	0.49	<0.001	<0.001
Creatinine Clearance (mL/min)	117.5	117.44	119	0.954	0.426	0.249

## Data Availability

Data is available at corresponding author main site upon reasonable reqeust.

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
