# Peer review of "Patient Blood Management in Microsurgical Procedures for Reconstructive Surgery"

_diagnostics, 2023, doi:10.3390/diagnostics13172758_

Round 1

Author Response

We thank the reviewer for the suggestions we took in deep consideration. In particular as regarding the first objection (clarifying the aim of the study) we now stated this more clearly in the introduction. As regarding the second suggestion (taking into consideration the invasiveness of the surgery) we now added some considerations were appropriate.  The structure of the manuscript is as well now respecting the given sections. 

Reviewer 2 Report

In this paper, the authors describe their management protocol for chronic inflammatory anemia for lower extremity reconstruction with free flaps. They found improvement in hematologic parameters, no need for blood transfusions, and no free flap necrosis.

There are major flaws in this study that require revision

Introduction – Too long. Reduce to 3-4 paragraphs

Introduction – Please state clearly the study’s objective and hypothesis in the introduction

Line 99-148 – delete. Redundant

Methods – please state clearly what are the primary and secondary outcomes studied.

Line 154 – Figure2  is unnecessary to this study

Line 165 – the tables belong in the results section

Line 173-174 – why did three patients receive something different? (iron gluconate). How was it decided which medicine they received?

Line 219 – have you used cell salvage techniques in the patients presented in this study? It is not mentioned in the results section

Discussion – what is the free flap failure rate in your institution? In the literature, it is around 2-5%. You have only included 16 patients in this study. With your hematologic protocol, none of your free flaps failed. Could it be that you just did not include enough?

There are several errors with English language. Please use an English correction service/translation

There are several English grammar mistakes, too many to count

English syntax requires improvement as well

Author Response

We thank the reviewer for the suggestions we took in deep consideration. In particular the introduction has now been shortenend accordingly to the suggestions.  The study hypothesis is now clearly stated right from the beginning.  The redundant part of the methods section now appears only there due to the reduction of the introduction.  The primary objective (blood transfusions) is now clarly stated in the methods paragraph, as well as secondary object (rejection of the limb) . As regarding figure 2 we respectfully disagree as it clearly helps the reader defining the context of the surgery. 

As regarding tables we divided them by prior or after intervention putting the baseline values in the methodology section but, as suggested, the resulting values in the results section. 

The reason for differential treatment (ferrocarboxymaltose vs iron gluconate) is now stated in the text.  No cell salvage techinque was used whatsoever and this is now stated in the text. The average free flap failure of the center is now indicated in the results section. We agree that the numerosity of the sample is a potential weakness of the study and we point it out in the text. 

Reviewer 3 Report

Dear Editor and Authors,

Thank you for the opportunity to review the manuscript entitled “Patient Blood Management in Microsurgical Procedures for Reconstructive Surgery.”. The authors aimed to o describe a therapeutic protocol with erythropoietic stimulating agents (ESAs) (Binocrit) and intravenous iron every other day, starting the second day after microsurgical surgery. They included 16 patients and reported an optimal therapeutic response. The topic is very interesting and the literature concerning this issue is scarce. However, I have some remarks:

-       Abstract: please add the aim of the paper

-       Add some more specific information to the Intro about blood loss during microsurgical procedures (e.g. rates of transfusions in other studies) – these data would justify your interest in the topic

-       I do not see any justification for Figure 1

-       The beginning of Methods should be moved to the Intro ! and at the end of this section you should state “In order to fill this gap in the literature we conducted a prospective observational study which aimed to…”

-       Methods: pls add more data about the patients’ condition: how long after injury, were they previously operated due to the injury, did you include also patients with multiple injuries or only those with lower limb fracture /complicated?/, what are “baseline”’ parameters /how long after primary injury/?

-       The used statistical tests should be presented in Statistical analysis rection in methods

-       Table 3 – T0,1,2 should be explained in the footnote

-       Proofreading /English editing/ is needed!

Author Response

We thank the reviewer for the suggestions we took in deep consideration. The aim of the study has now been clarified and stated from the beginning.  The literature evidence of blood trasnfusions in this surgical context has been clarified.  As suggested figure 1 has now been removed and incorporated into the text. 

As suggested part of the methods section has now been moved to introduction.  

The aim of the study has now been clearly stat ed at the end of the methods section as suggested.  

All patients were experiencing single and first trauma setting, this is now stated in the text.  

The description of the exact statistical text is now clarifed in the methodology section. 

As regarding table 3 the footnote now expalins t0 t1 and t2. 

Round 2

Reviewer 1 Report

The article has not reached the status for review. The reviewer recommends the author to receive the advice from his/her senior doctors what should be described in “Introduction”, “Material and Methods”, “Results”, and “Discussion”.

Author Response

We thank you for your careful considerations of this article.

We  revised and implemented the introduction , methods and results and conclusions. The statistic analysis was verified and we added the units measurement on all examined parameters.

Regarding the references , we have updated them , even if this is an innovative field.

Reviewer 2 Report

The authors replied to the requested major revisions, but some of these revisions are not included in the new manuscript submitted.

Please submit a point-by-point response to my previous revisions, with the location/lines in the manuscript where these revisions are made

Moderate editing required for English language

Author Response

We thank you for the careful considerations. We implemented and change the introduction, the methods , the results.

We verified the statistical analysis , and the research design.

Regarding the English language , we had the article revised by an experienced professional.

Round 3

Reviewer 1 Report

The reviewer is satisfied with the revision, and recommends being published.

Reviewer 2 Report

No further revisions

Minor English language mistakes